# Effect of prenatal alcohol consumption on dental enamel formation in offspring—An animal study protocol

**Roberta Duarte Leme, Guido Artemio Marañón-Vásquez, Juliana de Lima Gonçalves, Fabrício Kitazono de Carvalho, Alexandra Mussolino de Queiroz, Francisco Wanderley Garcia de Paula-Silva**⏺*

Department of Pediatric Dentistry, Ribeirão Preto School of Dentistry, University of São Paulo, Ribeirão Preto, SP, Brazil

* franciscogarcia@forp.usp.br

## Abstract

**Data Availability Statement:** No datasets were generated or analysed during the current study. All

### Background

The etiology of developmental defects of enamel (DDE) remains incompletely understood. Prenatal alcohol exposure has been proposed as a potential risk factor for DDE. Animal studies suggest that *in utero* ethanol exposure can disrupt ameloblast function, leading to enamel abnormalities. This study aims to: (1) Assess the impact of prenatal alcohol consumption on the clinical and structural properties of dental enamel in offspring; and (2) Investigate the underlying mechanisms of these alterations through histological and molecular analyses. Pregnant Wistar rats will be assigned to two groups: one exposed to ethanol and a control group with no alcohol exposure. Ethanol exposure will follow a binge drinking model, with rats receiving 3 g/kg of ethanol (30% w/v) for 3 consecutive days, followed by 4 days of rest each week. This regimen will begin one week prior to conception and continue throughout pregnancy. The incisors and molars of offspring will be evaluated on the 10th (n = 22 per group) and 28th (n = 22 per group) days of life. Visible enamel changes will be documented through photographs. Enamel volume, thickness, and density will be assessed using micro-CT imaging. Mechanical properties will be evaluated using the Knoop micro-hardness test, while chemical composition will be analyzed through Scanning Electron Microscopy with Energy Dispersive X-ray (SEM-EDX) and Raman spectroscopy, respectively. The area of the organic enamel matrix will be quantified in histological sections. Genes *Amelx*, *Enam*, *Ambn*, *Mmp2*, *Mmp9*, *Mmp20*, *Klk4*, *Cldn3*, *Cldn16*, and *Cldn19* will be evaluated in ameloblasts using real-time RT-PCR and protein synthesis will be confirmed by immunohistochemistry. Gelatinolytic activity in the ameloblast layer will be assessed by in situ zymography.

relevant data from this study will be made available upon study completion.

**Funding:** -Coordenação de Aperfeiçoamento de Pessoal de Nível Superior; Award Number: 001; Recipient: Roberta Duarte Leme. -Fundação de Amparo à Pesquisa do Estado de São Paulo; Award Number: 2023/12014-8; Recipient: Francisco Wanderley Garcia Paula-Silva. The funders had no role in study design, data collection and analysis, decision to publish, or preparation of the manuscript.

**Competing interests:** The authors have declared that no competing interests exist.

# Introduction

## Rationale of the study

Dental enamel formation, or amelogenesis, is a complex process orchestrated by ameloblasts [1]. These cells undergo a series of ultrastructural alterations to reach two major functional stages, secretory and maturation [2, 3]. While secretory-stage ameloblasts synthesize an enamel matrix template, maturation-stage ameloblasts promote enamel matrix mineralization [1, 3]. Disruptions that affect the function of these cells during the secretory or maturation stages of amelogenesis can lead to developmental defects in enamel (DDE), such as hypoplasia (*i.e.*, quantitative enamel defect) and hypomineralization (*i.e.*, qualitative enamel defect), respectively [4, 5].

The etiology of DDE has not been fully elucidated. Various prenatal, perinatal and postnatal conditions have been suggested as risk factors for DDE [6–9]. Prenatal factors can affect the enamel formation of deciduous teeth once it begins *in utero* during pregnancy [10, 11]. Prenatal exposure to undernutrition [10], hypertension [11], diabetes [12], use of anti-epileptic drugs [13], tobacco [11] and alcohol consumption [14] have been associated with DDE in the primary dentition.

Consumption of high amounts of alcohol during pregnancy can cause serious birth defects [15, 16]. Studies in mice embryos have shown that alcohol exposure at early stages of development (*i.e.*, gastrulation) reduces rates of mitotic activity and affects the migration process of mesodermal cells toward the primitive streak [17, 18]. Exposure to alcohol in animal embryos at this period results in death of neural crest cells destined to give rise to facial structures [19–21]. Dental alterations have been observed in offspring of mice exposed *in utero* to alcohol [22]. Delayed cell differentiation, reduced secretion of extracellular matrices, delayed calcification of the dentin matrix [23], and delayed tooth eruption [22, 24] have also been reported as consequences of alcohol intake during pregnancy in animal studies. Research in humans suggests that alcohol consumption during pregnancy is associated with an increased prevalence of DDE [25–27].

## State-of-the-art of alcohol effects on dental enamel development

Developing tooth enamel appears to be particularly susceptible to maternal alcohol consumption. Ultrastructural changes in secretory-stage ameloblasts were observed in the tooth germs of mini-pig fetuses after *in utero* ethanol exposure [28]. Ethanol consumption disrupts the activity of Epidermal Growth Factor (EGF) during odontogenesis, adversely affecting the formation of the enamel matrix in rodents [29, 30]. Consistent with these findings, a high incidence of enamel hypoplasia has been observed in children with Fetal Alcohol Syndrome [31]. Furthermore, it has been suggested that cells regulated by growth factors are particularly vulnerable to alcohol exposure [32]. Therefore, it is plausible to conclude that ameloblasts in the maturation stage, which are actively regulated by EGF, are similarly affected. Supporting this hypothesis, retrospective human studies have shown a correlation between prenatal alcohol consumption and an increased thickness of the neonatal enamel line. This suggests that alcohol exposure may induce physiological changes that disrupt calcium homeostasis during enamel deposition [14].

To further investigate the etiology of DDE (i) evaluating the effect of prenatal alcohol consumption on clinical and structural aspects of dental enamel in offspring and (ii) exploring the mechanisms of the possible alterations observed by gene and protein expression analyses of molecules involved in amelogenesis.

## Materials and methods

This protocol was approved by the Ethics Committee for Use of Animals of the School of Dentistry of Ribeirão Preto, University of São Paulo (FORP-USP), Brazil (# 2024.1.25.58.0). All experimental procedures will be conducted in accordance with the recommendations of the National Council for the Control of Animal Experimentation, Brazil, and the ARRIVE (Animal Research: Reporting of In Vivo Experiments) guidelines [33].

### Animals and experimental procedures

Eight-week-old Wistar rats (*Rattus norvegicus albinus*) weighing approximately 200g will be obtained from the Central Animal Facility of the University of São Paulo, campus of Ribeirão Preto. From then on, the animals will be kept at the FORP-USP Animal Facility in propylene cages with perforated stainless-steel lids, at constant temperature (22 ± 2˚C) and relative humidity (55 ± 10%), under a 12-hour light/dark cycle, and with standard laboratory diet and water *ad libitum*.

Exposure to alcohol will follow a binge drinking model, which is defined by the consumption of large amounts of alcohol within a short time frame, alternated with periods of abstinence [34–36]. This model was chosen because it more closely reflects contemporary patterns of alcohol use, providing a more accurate representation of real-world consumption. It contrasts with the chronic alcoholism model, which is characterized by significant impairment and drug dependence associated with persistent, excessive alcohol consumption [37], a pattern that is less common today. Various binge drinking models in rodents have been proposed [38–40]; however, protocols for application in pregnant animals are not completely established. The chosen binge drinking model should achieve the desired blood alcohol concentration (BAC) and be safe for the animals. According to the National Institute on Alcohol Abuse and Alcoholism (NIAAA), binge drinking is defined as raising blood alcohol concentration (BAC) to 0.08% or higher [41], a level that can induce signs of intoxication and impair performance on tasks involving timing, response inhibition, and position discrimination [42]. Some previous studies in rodents have shown that consumption of 3 g/kg of ethanol during pregnancy can lead to desired BACs of 150–200 mg/dL [43–45]. Specifically, this binge-drinking protocol implemented in Wistar rats for three consecutive days during pregnancy (6th, 7th and 8th gestational days) showed BACs of 177 ± 18.37 mg/dL with no reported animal deaths [44], appearing to be a valid and safe method.

For the present study, female rats will be randomly divided into a group exposed to ethanol consumption and a non-exposed control group. As previously described [44], the exposed animals will receive doses of ethanol via gavage (3g/Kg, [30% w/v]) for 3 consecutive days (first 3 days of the week) followed by 4 days of rest per week, to represent heavy and intermittent alcohol consumption (binge drinking model). To achieve a concentration of 30% ethanol, a dilution will be prepared using 30 ml of absolute alcohol mixed with 70 ml of distilled water. For dosing, the target of 3 g of ethanol per kg of body weight equates to 3.8 ml of ethanol per kg. Assuming each animal weighs approximately 200 g, the calculated dose will be 0.6 g of ethanol, corresponding to 0.76 ml of the prepared solution.

The exposed rats will begin receiving alcohol doses one week prior to mating. After this initial week, female rats from both groups will be paired with unexposed male rats (1:2 male-to-female ratio). Mating will last approximately five days [46], and pregnancy will be confirmed by the presence of sperm in the vaginal smear. If pregnancy does not occur within the expected time frame, the rat will be removed from the experiment to prevent potential alterations in the outcome due to prior alcohol exposure. Male rats used for mating will be euthanized once pregnancy is confirmed. The animals in the experimental group will continue alcohol exposure

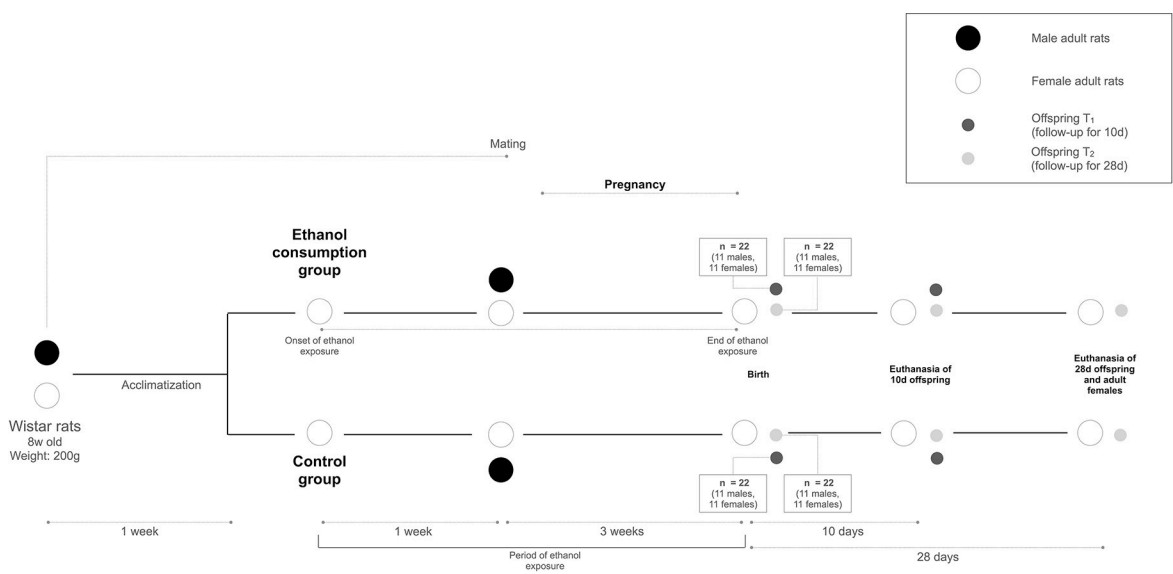

**Fig 1. Study design.**

throughout pregnancy, following the same protocol until the pups are born (approximately 21 days). Pregnant rats in the control group will receive distilled water. Female rats will be housed with their offspring for the duration of the study, with alcohol exposure lasting a total of 4 weeks.

The main study unit will be the offspring. Their incisors and molars will be evaluated at two times, on the $10^{th}$ ($T_1$, evaluation of newly erupted incisors and germs of first molars) and the $28^{th}$ ($T_2$, evaluation of incisors and molars in occlusion and function) day of birth. It is expected to evaluate a minimum of 20 offspring (10 males and 10 females) per group at each of these times, resulting in a sample size of 80 animals. Considering possible losses, this number will be increased by 10%, totaling 88 offspring required for the present study. Due to the absence of prior studies on this topic, there is insufficient data to calculate effect sizes for each assessed outcome, making a priori sample size estimation impossible; therefore, the power of the analyses will be calculated a posteriori. If necessary, mating procedures will be repeated until the planned sample size is reached, as well as a 1:1 male:female offspring ratio, so that the sample is more representative and appropriate for assessing possible sexual dimorphisms.

Adult rats will be euthanized in $CO_2$ chamber (flow 7 L/min), after having been anesthetized with Ketamine 100 mg/kg (Agener®, Agener União, São Paulo, SP, Brazil) and Xylazine 7.5 mg/kg (Syntec®, Syntec, Santana do Parnaíba, SP, Brazil) intraperitoneally. The offspring will be euthanized at the time of evaluations by anesthesia followed by decapitation. The study design is outlined in Fig 1.

## Animals monitoring

The weight of the pregnant rats will be evaluated at the $7^{th}$, $14^{th}$, and $21^{st}$ day of gestation. The survival rate of offspring during the study period will be assessed using the Kaplan-Meier estimator. Physical development of offspring will be monitored by weighing and measuring body length (cranial-caudal) at the $7^{th}$, $14^{th}$, $21^{st}$, and $28^{th}$ day after birth. The day of emergence of the lower incisors, as well as the opening of the eyes and ears, the initiation of solid food intake, and the weaning period, will also be recorded.

## Dental enamel evaluations

The incisors and first molars of the euthanized animals will be subjected to evaluations according to the distribution presented in Fig 2.

**Photography.** Clinically visible changes in dental enamel will be assessed through digital photographs taken after the animals' euthanasia. The photographs will be acquired with a digital single lens reflex camera (Eos Rebel T2i, Canon, Tokyo, Japan) attached to a 100 mm f/2.8 macro lens (Canon, Tokyo, Japan). The settings will be set to ISO 200, f32, 3s.

After 28 days, photographs will be taken of the incisors and molars (n = 22) from the right hemimaxilla of both the exposure and control groups. Additionally, incisors and molars (n = 22) from the left hemimaxilla of each group will also be photographed. For the mandible, photographs will be taken of the right hemimandible from both groups (n = 22), along with the incisors and molars (n = 22) from the left hemimandible of each group (Fig 2).

Photographs will be taken of the buccal surfaces of the incisors and the occlusal surfaces of the molars to document the presence and frequency of demarcated opacities, hypoplasias, and other developmental enamel defects (DDE). All animals will be positioned at a 90-degree angle to the camera lens. White light will be used for ambient lighting, and polarized light along with infrared filters will be employed to enhance the visualization of the enamel surface [47].

**Micro-computed tomography.** The volume, thickness, and density of the enamel in both incisors and molars will be assessed using micro-computed tomography (Micro-CT), as

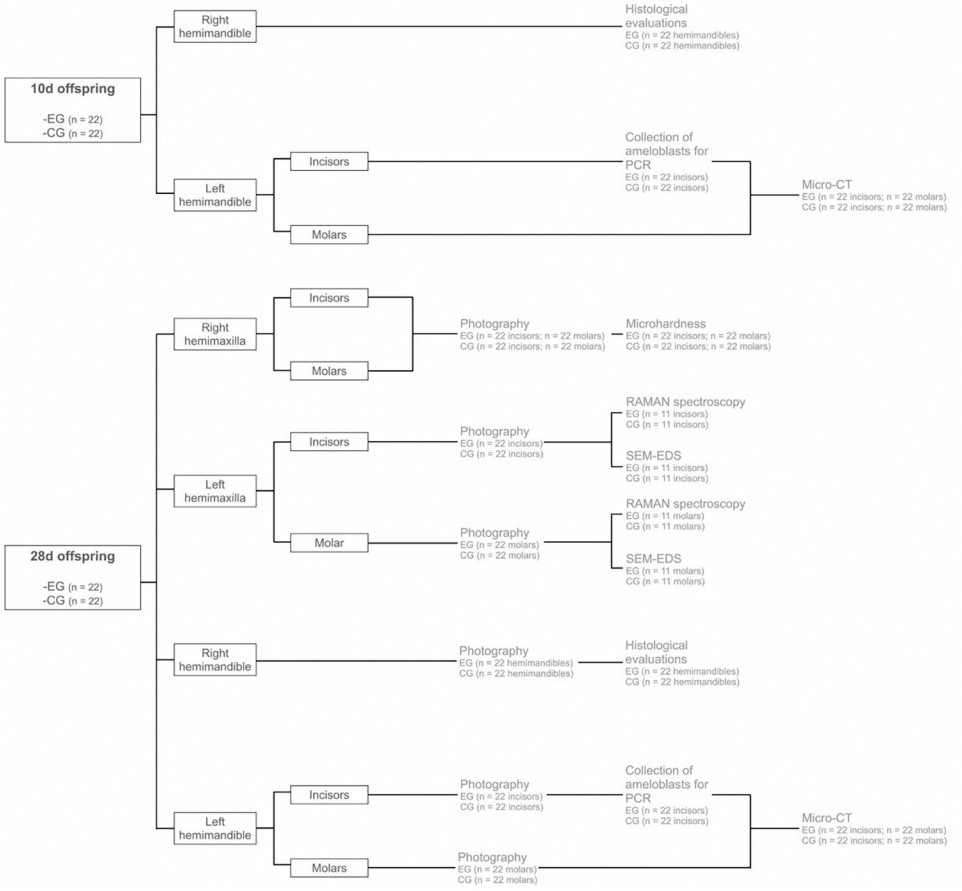

**Fig 2. Sample distribution and proposed analyses.** EG–Ethanol consumption group, CG–control group.

previously described [47]. At day 10, the analysis will be performed on incisors and molars (n = 22) from the left hemimandible of both the exposure and control groups. Similarly, at day 28, the analysis will be repeated on incisors and molars (n = 22) from the left hemimandible of each group (Fig 2). For incisors, the region of the tooth just above the bone ridge will be analyzed [48]. Prior to analysis, these teeth will be dissected to collect ameloblasts. Additionally, a preliminary Micro-CT scan will be conducted to ensure that the procedure does not affect cell viability.

Regarding image acquisition, a high-resolution desktop Micro-CT system (Phoenix V take xS240, GE, Boston, USA) will be used, operating under the following parameters: 70 kV, 200 µA, Al/Cu filter of 0. 1 mm, voxel size 5.4 µm, full circle rotation steps at 0.4˚ angle intervals, and average scanning time about 2 hours. To ensure the accuracy and standardization of the test, the same voltage, exposure time, and data analysis parameters will be applied to all samples.

Enamel volume ($mm^3$) will be evaluated using 3D Slicer 5.0.3 (http://www.slicer.org) and ITK-Snap 3.6.0 (http://www.itksnap.org) software. Enamel thickness and density will be measured in 2D projections from 3D scans in ImageJ software (Wayne Rasband, National Institutes of Health, USA).

**Knoop microhardness tes.** To detect changes in the mechanical resistance of the enamel, a microhardness testing machine (Shimadzu–HMV-2, Kyoto, Japan) will be used. This test will be conducted on incisors and molars (n = 22) from the right hemimaxilla of both the exposure and control groups at 28 days (Fig 2). Given the natural curvature of rodent teeth, the test will be conducted from a centralized point to create as flat a surface as possible.

A 10 gf load will be applied for 5 s using a Knoop diamond tip on the incisal edges, cusp tips, and the middle and cervical thirds of the buccal surfaces of the teeth. The average hardness (KHN) for each tooth will then be calculated based on these measurements.

**Scanning electron microscopy-energy dispersive X-ray.** Scanning electron microscopy (SEM) will be employed to assess morphological changes in the enamel structure. At day 28, incisors and molars (n = 11) from the left hemimaxilla of animals in both the exposure and control groups will be analyzed (Fig 2). Phosphoric acid will be used to etch the surface. The samples will be dried and treated with 37% phosphoric acid for 30 seconds, followed by washing with distilled water and a second drying.

The specimens will be fixed on stubs for SEM using double-sided adhesive carbon tape, then sputter-coated with gold in a vacuum metalizing machine (SDC 050; Bal-Tec AG, Balzers, Germany). They will be examined with a scanning electron microscope (JEOL JSM-6610LV). SEM images of the entire enamel thickness will be qualitatively evaluated at magnifications of 500x and 2000x. The rest of the hemissections will be used to evaluate the enamel mineral content by energy dispersive X-ray spectroscopy (EDS) using the Oxford Instruments INCA 300 EDX system (Abingdon, Oxfordshire, UK). The percentages of calcium (Ca), phosphorus (P), oxygen (O) and carbon (C) present in the incisal/occlusal, middle and cervical region of the teeth will be quantified.

**Raman spectroscopy.** The chemical composition of the enamel will be analyzed using Raman spectroscopy (Ocean Optics Spectrometer, Inc., Dunedin, FL, USA). Incisors and molars (n = 11) from the left hemimaxilla of animals in both the exposure and control groups will be evaluated at 28 days (Fig 2). Diode laser (λ = 785 nm) with spectral resolution of 11 $cm^{-1}$, excitation power of 400 mW, and five seconds of acquisition will be applied to the teeth. As previously described [49], three measurements will be taken from six distinct regions of each tooth. The average spectrum for each region will then be calculated based on these acquisitions. The spectra will be processed using the MatLab and the peaks selected for analysis will be those related to the phosphate presence; vibrational mode n1, n3 and n4 $(PO_3)^{-4}$ in

hydroxyapatite; symmetrical vibrational stretching of phosphate ions $[(PO_4)^{-3}]$; and carbonate presence. Additionally, the carbonate/phosphate ratio will be analyzed.

**Histological analysis.** The right hemimandibles from each group will be dissected at 10 and 28 days for subsequent analysis (n = 22) (Fig 2). The tissues will be fixed in 10% buffered formalin for 24 hours, demineralized in 5% ethylenediaminetetraacetic acid (EDTA, Merck/ Millipore, Burlington, MA, USA), and then processed using standard histological techniques.

**Histometric analysis.** H&E-stained histologic sections will be qualitatively evaluated by using conventional light microscopy (Zeiss Axiolab 5; Carl Zeiss AG Light Microscopy, Göttingen, Germany). For morphometric analysis, videomicroscopy performed with a Zeiss Axiocam 503 Color (Carl Zeiss AG Light Microscopy) at a magnification of 20x and x40 [50] will be used. The microscope will be operated in bright-field mode. For each sample, the area of the organic enamel matrix in um$^2$ will be outlined.

**Immunohistochemistry.** Immunohistochemical analysis will be performed for in situ identification of proteins in the ameloblast layer. The slides will be submitted to the recovery of antigenic epitopes using sodium citrate buffer solution pH 6.0, heated at 95˚C. Endogenous peroxidase will be blocked with 3% hydrogen peroxide for 40 minutes. Non-specific binding sites will be blocked with 5% bovine serum albumin (Sigma-Aldrich) for 60 minutes. The sections will be successively incubated with the primary antibodies for AMELX, ENAM, AMBN, MMP-2, MMP-9, MMP-20, KLK-4, CLAUDIN-3, CLAUDIN-16, AND CLAUDIN-19, at 4˚ C, overnight. Then, the sections will be washed and incubated with mouse anti-goat (sc- 2491, Santa Cruz Biotechnology) and mouse anti-rabbit (sc-2489, Santa Cruz Biotechnology) biotinylated secondary antibody for 1 hour, washed in phosphate-buffered saline (PBS), and incubated with streptavidin conjugated to horseradish peroxidase (HRP) for 20 minutes. A 3,3'-diaminobenzidine (DAB; Sigma-Aldrich) will be used as an enzyme substrate for 5 minutes. The sections will then be washed and counterstained with hematoxylin. Control slides, in which the primary antibody is omitted, will be included to assess the specificity of the immunostaining. The presence or absence of immunolabeling for the selected proteins will be evaluated using an Axiolab5 microscope coupled with an Axiocam MRc5 camera (Carl Zeiss). For quantification of immunostaining, ImageJ software (National Institutes of Health, Bethesda, MD) and the Color Deconvolution plugin will be employed. Data will be expressed as arbitrary units per μm$^2$.

***In situ* zymography.** The assessment of gelatinolytic activity in the ameloblast layer will be carried out by *in situ* zymography. The sections will be immersed in sodium borohydride (1 mg/ml; Sigma) for 15 minutes (3x), washed in PBS and incubated with a gelatinous substrate linked to fluorescein isothiocyanate (DQTM Gelatin, Molecular Probes, Eugene, USA) dissolved in agarose (0.1 mg/ml; Sigma), for 2 h at 37˚ C, in a dark humidified chamber. The labeling of the DNA present in the cell nuclei will be carried out with 4'-6-diamidino-2-phenylindole (DAPI; 0.5 μg/ml) added to the incubation medium. Control slides will be pre-incubated in 20 mM ethylenediaminetetraacetic acid (EDTA, Sigma) for 1 hour, after which EDTA will be added to the incubation medium. The quantification of gelatinolytic activity in the slides will be performed using fluorescence microscopy. Fluorescence spots in representative areas of the sections will be counted under 10x magnification and the data will be expressed as the number of fluorescence spots per mm$^2$.

**Real time reverse transcription-polymerase chain reaction.** The expression of mRNA encoding proteins involved in amelogenesis will be evaluated by real-time reverse transcription-polymerase chain reaction (RT-PCR). Enamel organs rich in ameloblasts will be collected from the lower incisors (n = 22) of both the exposure and control groups (Fig 2), following a previously established protocol [51]. The incisors will be carefully isolated from the mid-mandible. Next, a brief cleaning will be conducted using phosphate-buffered saline (PBS 1X),

followed by scraping the surface corresponding to the ameloblasts in either the secretion or maturation phase, using an excavator or a similar tool. Finally, the cervical loop of the initiator will be removed from the apical portion. Total RNA from these cells will be obtained by extraction using the PureLink RNA Mini Kit (Invitrogen, Thermo Fisher Scientific Inc, Wilmington, NC). Following the above-mentioned protocol, it is expected to achieve about 5–10 µg total RNAs for each preparation [51]. The cDNA will be synthesized by a reverse transcription reaction, starting with 1 mg of total RNA (High-Capacity cDNA Reverse Transcription Kit; Applied Biosystems, Foster City, CA). The genes selected for investigation will be *Amelx*, *Enam*, *Ambn*, *Mmp2*, *Mmp9*, *Mmp20*, *Klk4*, *Cldn3*, *Cldn16*, and *Cldn19*. The genes for the enzyme glyceralde-hyde-3-phosphate dehydrogenase (*Gapdh*, Mm99999915_g1) and beta-actin (*Actb*, Mm02619580_g1) will be used as reference. Amplification will be performed initially at 95°C for 2 seconds, followed by 40 cycles at 95°C for 1 second and at 60°C for 20 seconds. Relative quantification of gene expression will be performed using the ΔΔCt method [52].

## Statistical analysis

Data will be analyzed using GraphPad Prism 8.0 (Prism, Chicago, IL, USA), with a significance threshold set at 5%. The relative and absolute frequencies of clinical enamel changes identified in photographic analyses, as well as enamel mineral content assessed by SEM-EDS in both groups, will be compared using the chi-square test. Continuous data from microhardness assessments and Raman spectroscopy will be analyzed using one-way ANOVA to determine the effects of alcohol. Data from microtomographic analyses (including enamel volume, thickness, and density), histomorphometric analysis (enamel matrix area), immunohistochemistry, in situ zymography, and PCR (relative mRNA expression) will be evaluated using two-way ANOVA to investigate the effects of alcohol, age, and their interaction. Separate analyses will be performed for incisors and molars. If the assumptions for these tests are not met, generalized models will be used based on the characteristics of the data.

Photographic evaluations will be conducted by two evaluators, and the Kappa correlation coefficient will be calculated to measure their agreement. The results of the Micro-CT and histometric analysis will be evaluated on two separate occasions, two weeks apart, to determine method error. Intra-evaluator repeatability will be measured using the Intraclass Correlation Coefficient, while the presence of random and systematic errors will be examined using the Bland-Altman method. In the analyses of Knoop microhardness, Raman spectroscopy and energy-dispersive X-ray scanning electron microscopy, all samples will be discarded after completion of the tests. Finally, in histological analyses, routine procedures will be conducted with the utmost care and precision to the greatest extent possible.

## Expected results

This research will provide insights into how adverse effects during pregnancy can influence the formation and biomineralization of dental enamel in an animal model. The findings may enhance our understanding of the impact of prenatal alcohol exposure on the development of dental enamel defects (DDE) and clarify the mechanisms of amelogenesis that are altered by this exposure. The presence of columnar ameloblasts and odontoblasts is anticipated, along with a reduction in the thickness of both enamel and dentin [53]. Additionally, it is likely that the size of the tooth germ will be diminished. Macroscopic analyses are expected to reveal lesions in the enamel, characterized as areas of hypomineralization or opacities. Consequently, a decline in mineral content is anticipated, which will directly affect the microhardness and density of the tissue, as well as reduce the expression of key peptides involved in enamel formation and maturation.

For the SEM-EDS, RT-RAMAN, and in situ zymography outcomes, due to the absence of prior studies, these are considered exploratory analyses, and the expected results cannot be precisely defined. Additionally, the methods employed for each procedure, such as ameloblast collection, may be refined or adjusted as the research advances. Ultimately, the findings from this study aim to inform strategies for enhancing maternal and child oral health, promoting health education, and improving prenatal dental care, thereby contributing to better quality of life and overall well-being for individuals.

## Author Contributions

**Conceptualization:** Roberta Duarte Leme, Fabrício Kitazono de Carvalho, Alexandra Mussolino de Queiroz, Francisco Wanderley Garcia de Paula-Silva.

**Funding acquisition:** Francisco Wanderley Garcia de Paula-Silva.

**Methodology:** Roberta Duarte Leme, Guido Artemio Marañón-Vásquez, Juliana de Lima Gonçalves, Fabrício Kitazono de Carvalho, Alexandra Mussolino de Queiroz, Francisco Wanderley Garcia de Paula-Silva.

**Supervision:** Francisco Wanderley Garcia de Paula-Silva.

**Writing – original draft:** Roberta Duarte Leme, Guido Artemio Marañón-Vásquez.

**Writing – review & editing:** Juliana de Lima Gonçalves, Fabrício Kitazono de Carvalho, Alexandra Mussolino de Queiroz, Francisco Wanderley Garcia de Paula-Silva.

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
