## [Decision Letter · Decision Letter 0]

2 Jun 2024

PONE-D-24-17362Effect of prenatal alcohol consumption on dental enamel formation in offspring – an animal study protocolPLOS ONE

Dear Dr. de Paula-Silva,

Thank you for submitting your manuscript to PLOS ONE. After careful consideration, we feel that it has merit but does not fully meet PLOS ONE’s publication criteria as it currently stands. Therefore, we invite you to submit a revised version of the manuscript that addresses the points raised during the review process.

We look forward to receiving your revised manuscript.

Kind regards,

Boyen Huang, DDS, MHA, PhD

Academic Editor

PLOS ONE

Journal Requirements:

4. We note that Figure 1 in your submission contain copyrighted images. All PLOS content is published under the Creative Commons Attribution License (CC BY 4.0), which means that the manuscript, images, and Supporting Information files will be freely available online, and any third party is permitted to access, download, copy, distribute, and use these materials in any way, even commercially, with proper attribution. For more information, see our copyright guidelines: http://journals.plos.org/plosone/s/licenses-and-copyright.

Reviewers' comments:

Reviewer's Responses to Questions

**Comments to the Author**

1. Does the manuscript provide a valid rationale for the proposed study, with clearly identified and justified research questions?

Reviewer #1: Yes

Reviewer #2: Partly

2. Is the protocol technically sound and planned in a manner that will lead to a meaningful outcome and allow testing the stated hypotheses?

Reviewer #1: Yes

Reviewer #2: Partly

3. Is the methodology feasible and described in sufficient detail to allow the work to be replicable?

Reviewer #1: Yes

Reviewer #2: Yes

4. Have the authors described where all data underlying the findings will be made available when the study is complete?

Reviewer #1: No

Reviewer #2: Yes

5. Is the manuscript presented in an intelligible fashion and written in standard English?

Reviewer #1: Yes

Reviewer #2: Yes

6. Review Comments to the Author

You may also provide optional suggestions and comments to authors that they might find helpful in planning their study.

Reviewer #1: PONE-D-24-17362

The article entitled “Effect of prenatal alcohol consumption on dental enamel formation in offspring – an animal study protocol" by Leme et al. proposed to analyze the rat tooth enamel development with the embryonic alcohol exposure. This is a novel research area which tried to investigate the effect of alcohol on causing developmental defects of the enamel.

They are using various methods to investigate the structure of the enamel and tried to identify the underlying protein gene activity in causing these enamel defects.

I really enjoyed reading the proposed project and I have following comments to improve the content of this protocol.

1. Under the animals and experimental procedures

Exposure to alcohol will be carried out following a binge drinking model, in which the animals will receive doses of ethanol via gavage (3g/Kg, [30% w/v]) for 3 consecutive days followed by 4 days of rest per week [33, 34]. The animals of the control group will receive distilled water. After one week of ethanol exposure ….

Explain the rationale of using this alcohol concentration. The cited two references have similar concentrations what is the rationale for using the (30% W/V) versus 1- 5% of alcohol concentrations.

2. The figure 2 has a confusion on left hemi mandible – Are you going to use the incisors and molars both for Rt-PCR and micro-CT? How you are going to do that? Please explain.

3. In the histological evaluations under histometric analysis “Microscope will be operated in fluorescence mode” Why you need the fluorescence mode as these sections will be H &E sections.

4. Please indicate the data availability according to the journal policy.

Reviewer #2: Roberta Duarte Leme and co-workers propose a binge drinking experimental model to study the effects of 3g/Kg alcohol [30% w/v] in pregnant rats on dental development of the pups.

The introduction part should be re-organized:

1- rationale of the study

2- State-of-the-art of alcohol effects on dental development

The paragraph on general effects of alcohol on embryogenesis should be reduced as it is out of the scope

References should be up-dated by avoiding papers written in non-English language

Authors should distinguish studies on humans and on animals

Figure 1: authors should add the time of exposure to alcohol

Animals: Doses, time and duration of exposure should be justified

Overall, investigations of 10-days-old rats (PND 10) should be revised to the fact that neither incisors nor molars are fully erupted

Photography: not sure that authors could see some defects on incisal enamel due to the very short time of adversity and incomplete development at PND 10

Knoop microhardness test: it would be more appropriate to investigate enamel hardness by nano-indentation rather micro-indentation due to the small size of samples

Histology: Authors should justify why “bone blocks containing incisives and molars will be removed from the previously dissected mandibles and maxillae.”

Real time reverse transcription-polymerase chain reaction (RT-PCR): not sure that authors will have enough material for RT-PCR analysis especially at PND 10

Target genes and proteins: authors should add MMP2 and MMP9 to be able to compare their data with zymography

Figure 2 should be corrected in accordance with modifications of the protocol

Is it possible to add a paragraph on expected data?

Abstract should be revised according to modifications of the paper

Reference list should be completely revised. Main studies reporting alcohol effects on dental development are quiet old, authors should up-date the reference list and include recent papers (de Carvalho P, et al., 2022; Blanck-Lubarsch M et al., 2019; etc…). They should add the important review of Silva MJ et al., 2016 reporting an association between alcohol consumption and MIH.

7. PLOS authors have the option to publish the peer review history of their article (what does this mean?). If published, this will include your full peer review and any attached files.

Reviewer #1: **Yes: **Dr. Devi Atukorallaya

Reviewer #2: **Yes: **Sylvie Babajko

---

## [Author Response · Author response to Decision Letter 0]

13 Jul 2024

July 12th, 2024

Dr. Boyen Huang, DDS, MHA, PhD

Academic Editor

PLOS ONE

PONE-D-24-17362

Effect of prenatal alcohol consumption on dental enamel formation in offspring – an animal study protocol

Dear Editor,

Thank you very much for your message. We are submitting a revised version of our manuscript after addressing the areas that required clarification, as mentioned by the reviewers. We appreciate the valuable suggestions to our manuscript. Additional information has been added/suppressed/modified as requested.

Please find our answers below. We hope our corrections are appropriate and that the manuscript may be reconsidered for publication. Please let us know if you have any questions.

Thank you very much for your consideration.

Warm regards,

The Corresponding author and co-workers

Reviewer comments

Reviewer #1 - Dr. Devi Atukorallaya

The article entitled “Effect of prenatal alcohol consumption on dental enamel formation in offspring – an animal study protocol" by Leme et al. proposed to analyze the rat tooth enamel development with the embryonic alcohol exposure. This is a novel research area which tried to investigate the effect of alcohol on causing developmental defects of the enamel.

They are using various methods to investigate the structure of the enamel and tried to identify the underlying protein gene activity in causing these enamel defects.

I really enjoyed reading the proposed project and I have following comments to improve the content of this protocol.

1. Under the animals and experimental procedures

Exposure to alcohol will be carried out following a binge drinking model, in which the animals will receive doses of ethanol via gavage (3g/Kg, [30% w/v]) for 3 consecutive days followed by 4 days of rest per week [33, 34]. The animals of the control group will receive distilled water. After one week of ethanol exposure ….

Explain the rationale of using this alcohol concentration. The cited two references have similar concentrations what is the rationale for using the (30% W/V) versus 1- 5% of alcohol concentrations.

Authors: Thank you very much for the positive evaluation of our manuscript. Below are the responses to your comments.

We are providing the rationale for the binge drinking model used (please see the 2nd and 3rd paragraphs of the 'Animals and experimental procedures' section). We decided to use the indicated alcohol dose based on previous literature, with the objective of achieving the blood alcohol concentration that characterizes a heavy/binge drinking pattern.

2. The figure 2 has a confusion on left hemi mandible – Are you going to use the incisors and molars both for Rt-PCR and micro-CT? How you are going to do that? Please explain.

Authors: Thanks for your comment. We plan to perform Rt-PCR and micro-CT, both on the lower incisors. For evaluation with micro-CT, the part of the tooth just above the bone ridge will be sectioned, to perform measurements previously made by our team (Ref. 44 of the manuscript). The intraosseous section of the tooth will be used to collect ameloblasts following a previously published protocol (Ref. 46 of the manuscript). The molars, as indicated in Fig 2, will only be used for micro tomographic evaluations.

We added information to the 'Micro-computed tomography' and 'Real-time reverse transcription polymerase chain reaction' sections for better understanding of the readers.

3. In the histological evaluations under histometric analysis “Microscope will be operated in fluorescence mode” Why you need the fluorescence mode as these sections will be H &E sections.

Authors: Thank you for your comment. We apologize for the mistake. In fact, in this case it is not necessary to use the microscope in fluorescence mode. Bright-field microscopy will be used for these analyses. This information was corrected in the 'Histometric analysis' section.

4. Please indicate the data availability according to the journal policy.

Authors: Thank you very much for your comment. We review the journal's data availability policy. Follow what is indicated:

What if my article does not contain any data?

All articles must include a Data Availability Statement but some submissions, such as Registered Report Protocols and Lab or Study Protocol articles, may not contain data. For manuscripts that do not report data, authors must state in their Data Availability Statement that their article does not report data and the data availability policy is not applicable to their article.

(https://journals.plos.org/plosone/s/data-availability)

Therefore, we stated the following: ''This article does not report data and the data availability policy is not applicable".

Reviewer #2 – Dr. Sylvie Babajko

Roberta Duarte Leme and co-workers propose a binge drinking experimental model to study the effects of 3g/Kg alcohol [30% w/v] in pregnant rats on dental development of the pups.

The introduction part should be re-organized:

1- rationale of the study

2- State-of-the-art of alcohol effects on dental development. 

The paragraph on general effects of alcohol on embryogenesis should be reduced as it is out of the scope.

References should be up-dated by avoiding papers written in non-English language.

Authors should distinguish studies on humans and on animals.

Authors: Thank you very much for your valuable appreciations. Below are our responses to your comments.

We re-organized the introduction as suggested. Although we reduced the paragraph on general effects of alcohol on embryogenesis, we decided to maintain some information as we consider it provides an adequate background/rationale for the present study. The references were updated, using suggestions from articles recommended by you, and we avoided the use of non-English language manuscripts. We appropriately indicate in which sample each referenced study was carried out (in humans or in experimental animal models).

Figure 1: authors should add the time of exposure to alcohol.

Authors: Thank you for your comment. The time of exposure to alcohol was added in Fig 1.

Animals: Doses, time and duration of exposure should be justified.

Authors: Thank you for your comment. The rationale for the alcohol exposure protocol used was provided (please see the 2nd and 3rd paragraphs of the 'Animals and experimental procedures' section).

Overall, investigations of 10-days-old rats (PND 10) should be revised to the fact that. Neither incisors nor molars are fully erupted.

Authors: Thanks for your comment. We agree that both teeth have not fully erupted in 10-day-old rats. In fact, the molars are still developing intrabony during this phase.

The incisors emerge through the gum between 8 and 10 days after birth (Addison and Appleton, 1915; Schour and Massler, 1949), while the molars do so around day 19 (Schour and Massler). At 28 days, approximately one week after weaning, the first and second molars are already in occlusion for a few days (Denes et al., 2018).

Our objective in choosing the evaluation moments indicated in the protocol is to evaluate the enamel in its different stages: pre-eruptive (molars at 10 days), eruptive (incisor at 10 days) and post-eruptive and in function (incisors and molars at 28 days).

Specifications were added in the fourth paragraph of the 'Animals and experimental procedures' section.

Photography: not sure that authors could see some defects on incisal enamel due to the very short time of adversity and incomplete development at PND 10.

Authors: Thank you for your comment. We agree with your comment. However, we chose to maintain these evaluations in the protocol for clinical recording of the enamel of recently erupted incisors (see response to the previous comment).

Knoop microhardness test: it would be more appropriate to investigate enamel hardness by nano-indentation rather micro-indentation due to the small size of samples.

Authors: Thanks for your comment. Although we understand your suggestion, for logistical reasons, we have chosen to maintain enamel microhardness evaluations using the Knoop microhardness test. Despite the small size of the samples, our team has already successfully carried out the evaluation of the microhardness of the enamel of rodent incisors with this method (Ref. 44 of the manuscript).

Histology: Authors should justify why “bone blocks containing incisives and molars will be removed from the previously dissected mandibles and maxillae.”

Authors: Thanks for your observation. We made some modifications in this regard in the manuscript. The phrase mentioned (section 'Histometric analysis') makes sense for histological evaluations (i.e., histometric analysis, immunohistochemistry and in situ zymography). In this case, the incisors and molars will not be separated from their surrounding bone to avoid damaging the enamel that will be evaluated.

In the case of Rt-PCR evaluations, which will be performed only on lower incisors, we will follow a previously reported protocol for collecting ameloblasts (Ref. 46 of the manuscript); therefore, the teeth will not be removed with their surrounding bone. This last specification was added in the 'Real time reverse transcription-polymerase chain reaction' section.

Real time reverse transcription-polymerase chain reaction (RT-PCR): not sure that authors will have enough material for RT-PCR analysis especially at PND 10.

Authors: Thank you for your important observation. We will follow a previously published protocol (Ref. 46 of the manuscript) for appropriate collection of ameloblasts and obtaining enough total RNA for the analyses. We already have performed pilot studies in our laboratory and could collect enough amount of total RNA to evaluate gene expression.

This information was added in the 'Real time reverse transcription-polymerase chain reaction' section. Considering that at PND 10 the lower incisors will have already emerged into the oral cavity, we assume that ameloblast collection, following the indicated protocol, will be possible.

Target genes and proteins: authors should add MMP2 and MMP9 to be able to compare their data with zymography

Authors: Thank you for your important observation. MMP2 and MMP9 were added as target proteins and genes for immunohistochemistry and Rt-PCR analyses, respectively (please see sections 'Immunohistochemistry' and 'Real time Reverse Transcription-polymerase chain reaction').

Figure 2 should be corrected in accordance with modifications of the protocol.

Authors: Despite the modifications made according to the suggestions, we consider that the protocol for outcome evaluations will be maintained. Therefore, Fig 2 was not modified.

Is it possible to add a paragraph on expected data?

Authors: We added a paragraph on expected results (please see 'Expected results' section after the 'Statistical analysis' section).

Abstract should be revised according to modifications of the paper.

Authors: The abstract was revised according to the modifications made in the paper.

Reference list should be completely revised. Main studies reporting alcohol effects on dental development are quite old, authors should up-date the reference list and include recent papers (de Carvalho P, et al., 2022; Blanck-Lubarsch M et al., 2019; etc…). They should add the important review of Silva MJ et al., 2016 reporting an association between alcohol consumption and MIH.

Authors: The reference list was revised. Recent papers on the alcohol effects on dental development were added as suggested.

---

## [Decision Letter · Decision Letter 1]

20 Aug 2024

PONE-D-24-17362R1Effect of prenatal alcohol consumption on dental enamel formation in offspring – an animal study protocolPLOS ONE

Dear Dr. de Paula-Silva,

Thank you for submitting your manuscript to PLOS ONE. After careful consideration, we feel that it has merit but does not fully meet PLOS ONE’s publication criteria as it currently stands. Therefore, we invite you to submit a revised version of the manuscript that addresses the points raised during the review process.

We look forward to receiving your revised manuscript.

Kind regards,

Boyen Huang, DDS, MHA, PhD

Academic Editor

PLOS ONE

Reviewers' comments:

Reviewer's Responses to Questions

**Comments to the Author**

1. Does the manuscript provide a valid rationale for the proposed study, with clearly identified and justified research questions?

Reviewer #2: Yes

Reviewer #3: Partly

Reviewer #4: Partly

2. Is the protocol technically sound and planned in a manner that will lead to a meaningful outcome and allow testing the stated hypotheses?

Reviewer #2: Yes

Reviewer #3: Partly

Reviewer #4: No

3. Is the methodology feasible and described in sufficient detail to allow the work to be replicable?

Reviewer #2: Yes

Reviewer #3: No

Reviewer #4: No

4. Have the authors described where all data underlying the findings will be made available when the study is complete?

Reviewer #2: Yes

Reviewer #3: No

Reviewer #4: No

5. Is the manuscript presented in an intelligible fashion and written in standard English?

Reviewer #2: Yes

Reviewer #3: Yes

Reviewer #4: Yes

6. Review Comments to the Author

You may also provide optional suggestions and comments to authors that they might find helpful in planning their study.

Reviewer #2: Despite we may still discuss the feasibility of all investigations at PND10, authors replied to most of the comments, did preliminary pilot studies for feasibility and improved their manuscript.

Reviewer #3: This is a study protocol to examine the effect of prenatal alcohol consumption on dental enamel formation using rats as the model.

The study's scope is highly relevant to understanding the effect of prenatal maternal alcohol consumption on dental enamel formation. In humans, the enamel matrix of the primary teeth is laid down in utero, and matrix maturation also starts in utero. Unlike primary teeth, the majority of permanent teeth mineralization happens postnatally. Therefore, the effect of prenatal alcohol drinking is studied on populations with primary dentition or mixed dentition.

In rodents like rats and mice, the first molar and incisor enamel organs are formed prenatally; however, unlike human teeth, the enamel matrix mineralization happens postnatally in all teeth.

Therefore, this study design raises a fundamental question regarding the period of alcohol treatment for the mother mice: Is the prenatal alcohol treatment reasonable enough to observe the effect on the enamel matrix formation and mineralization, which happens postnatally? Further, how can the outcomes be applied to human tooth formation?

Overall comments on the methodology:

• The detail level is inconsistent method by method.

• Too many sites refer to previous reports about the methodology. The authors need to present all the steps of the procedure.

• All the experiments need to contain fundamental information, i.e., what types of specimens were used and how they were prepared, the number of samples, etc.

• There is no mention of the sample size in each experiment.

• The resolution of the two figures is too low to read.

The following are the section-by-section comments.

Animal and experimental procedure:

• There is no mention of controlling and monitoring water and chow consumption.

• The rationale for the sample size is not clear. How it's determined?

• The sex of the offspring should be accounted for.

Micro-computed tomography:

• The meaning of the sentence “ The region of the molars and surrounding bone will be removed for analysis” is unclear. Does it mean bone and molars will be dissected out from the mandible? If so, why is it necessary? Analyzing software can extract the preferred target region on the microCT image.

Scanning electron microscopy-energy dispersive X-ray

• What kind of instrument will be used to cut the tooth?

• How can artifacts caused by the cutting process be avoided?

• Any plan to etch the surface?

Histometric analysis

• The meaning of the sentence, “Bone blocks containing incisives (incisors?) and molars will be removed from the previously dissected mandibles and maxillae.”, is unclear.

• There is no description of what kind of structure will be observed after H&E staining.

• Magnification x10 would be too small to observe cells and matrix structures.

In situ zymography

• The procedure how to obtained the section was missing

Real time reverse transcription-polymerase chain reaction

• The collection of the ameloblasts is not described well enough.

• The methods in reference #46 are not for the pure ameloblasts collection but for enamel organs enriched with ameloblasts. Authors need to be aware of it and note it.

Expected results

• Compared to the amount of the experiment information, expected results are too limited. Some reports use a similar method to this protocol (reference #29, 30, and DOI: 10.2485/jhtb.16.61, etc.). Therefore, the authors should be able to present the expected results of each experiment based on the previous reports.

Reviewer #4: This paper highlights an important area of research addressing the effects of maternal alcohol consumption on the development of enamel. Thank you to the authors for putting this paper together and working towards helping us better understand the impact of prenatal alcohol consumption on developing dentition. Some overall concerns raised by this paper are listed below:

There are multiple tests completed in this study, with many outcome evaluations completed—photography, micro-computed tomography, Knoop microhardness test, scanning electron microscopy-energy dispersive x-ray, Raman spectroscopy, histometric analysis, immunohistochemistry, in situ zymography, real time reverse transcription-polymerase chain reaction, and statistical analysis also. It is inferred that the developmental defects of enamel (DDE) will be found in offspring from rats exposed to alcohol consumption compared to those without. With 10 different DDE analysis techniques employed, no hypothesis was provided for any of the 10 tests. Without clarity on the reasoning and anticipated findings of these tests, the study lacks focus and does not have clearly defined outcomes. It seems that the tests have been assembled in an exploratory manner without a specific, well-defined objective. A ‘Study Protocol’ should have clearly defined outcomes for the physical properties (volume, thickness and density) of enamel and cellular changes associated with alcohol consumption.

Another concern is that the proposal does not have any content on standardising the tests or assessing the error. Each test described in the paper needs more detailed elaboration.

Power calculation should be an essential part of ‘Study Protocol’.

Statistical testing: The description on statistical testing is inadequate. Each quantitative method should contain adequate description for statistical testing. The authors should also discuss what they will do if the underlying assumptions of ANOVA or similar tests are violated.

Additional comments:

Animals and experimental procedures:

- How long is the mating period? Are female rats exposed to alcohol throughout this period?

- How long will the rats be exposed to alcohol in total? What is done if the female rat does not become pregnant in the expected mating period?

- “Binge drinking is characterised by bringing BAC up to 0.08 grams % or above, having the potential to generate signs of intoxication” – what are the signs of intoxication?

- Please clarify for a general reader why was a binge drinking model chosen and not a lower alcohol consumption percentage.

- P7: 3g/kg (i.e., 3gm/1,000gm x 100%) yields a value of 0.3%, and it is difficult to determine how this equates to 30% w/v.

The specific issues with some of the methods are provided as examples below.

Photography:

- Some of the key data are missing, e.g., camera angulation, room lighting, magnification, and quantitative vs qualitative analysis.

Micro-computed tomography:

- How will the authors ensure the density measurement is accurate and standardised? Will the molars and surrounding bone be removed from the micro-CT images or from the specimens before imaging?

- “… the collected incisors and molars will be mesiodistally hemisected and one of those sections will be prepared for SEM” – was there uniformity with the selected hemisection? How will this be selected and how will the orientation be standardised?

Knoop microhardness test:

- Will this test be carried out on flat or curved surfaces? What will be the orientation of the specimens/sections?

- Figure 2: For 10d offspring, microhardness testing will only occur for incisors and not molars. Although it is noted in the text that the molars are still under development, the figure should stand independent to the text, and this is not made clear for the general reader. There are similar issues with other tests.

Scanning electron microscopy-energy dispersive x-ray:

- What many spots will be examined from each specimen? Will there be outputs for other elements?

7. PLOS authors have the option to publish the peer review history of their article (what does this mean?). If published, this will include your full peer review and any attached files.

Reviewer #2: No

Reviewer #3: No

Reviewer #4: No

---

## [Author Response · Author response to Decision Letter 1]

26 Oct 2024

Reviewer comments

Reviewer #3:

Overall comments on the methodology

The detail level is inconsistent method by method.

Too many sites refer to previous reports about the methodology. The authors need to present all the steps of the procedure.

All the experiments need to contain fundamental information, i.e., what types of specimens were used and how they were prepared, the number of samples, etc.

Authors: Thank you for your comment. The methodology section has been revised to detail and reinforce the conditions for reproducibility. All updates are highlighted in red.

There is no mention of the sample size in each experiment.

Authors: Thank you for your comment. The sample size for each experiment has been included in Figure 2.

The resolution of the two figures is too low to read.

Authors: Thank you for your comment. The images were already adjusted and verified to comply with the submission requirements, at a resolution of 600 dpi.

Animal and experimental procedure:

There is no mention of controlling and monitoring water and chow consumption.

Authors: Thank you for your comment. The animals will be subjected to a standard laboratory diet with free access to water and feed. We have added this information to the methodology section to make it clearer and more detailed.

The rationale for the sample size is not clear. How it's determined?

Authors: Thank you for your comment. We decided to work with an estimate of 10 offspring of each sex per group. Taking potential losses into account, this amounts to a total of 22 offspring per group. Due to the absence of prior studies on this topic, we do not have data to calculate effect sizes for each assessed outcome, making it impossible to conduct a priori sample size estimation. In this proposal, we have opted to work with 88 animals and will calculate the power of the analyses a posteriori.

The sex of the offspring should be accounted for.

Authors: Thank you very much for your comment. We plan to conduct the counting of males and females so that the sample is more representative, with the appropriate ratio for assessing possible sexual dimorphisms. This information has been added to the text for better understanding and is highlighted in red.

Micro-computed tomography

The meaning of the sentence “The region of the molars and surrounding bone will be removed for analysis” is unclear. Does it mean bone and molars will be dissected out from the mandible? If so, why is it necessary? Analyzing software can extract the preferred target region on the Micro-CT image.

Authors: Thank you for your comment. The incisor must be dissected to collect ameloblasts beforehand. Additionally, we decided against performing Micro-CT prior to separating the pieces to ensure that this procedure does not affect the viability of the cells.

Scanning electron microscopy-energy dispersive X-ray

What kind of instrument will be used to cut the tooth? How can artifacts caused by the cutting process be avoided?

Authors: Thank you for your valuable consideration. The incisor hemisection strategy will not be implemented. We will analyze the surface of enamel.

Any plan to etch the surface?

Authors: Yes, we will use phosphoric acid to etch the surface. The samples will be dried and treated with 37% phosphoric acid for 30 seconds, followed by washing with distilled water and a second drying.

Histometric analysis

The meaning of the sentence, “Bone blocks containing incisives (incisors?) and molars will be removed from the previously dissected mandibles and maxillae.”, is unclear.

Authors: Thank you for your comment. In fact, only the right hemimandibles will be used for subsequent histological processing. We have replaced any potentially confusing terms, and all updates are highlighted in red.

There is no description of what kind of structure will be observed after H&E staining.

Authors: Thanks for your comment. The H&E staining will be used to observe ameloblasts.

Magnification x10 would be too small to observe cells and matrix structures.

Authors: Thank you for your suggestion. Reviewing studies that evaluated ameloblasts in Wistar rats, we found that it may be necessary to use higher magnification levels, such as 20 and 40x. This information has been added to the text to enhance clarity.

In situ zymography

The procedure how to obtain the section was missing.

Authors: Thank you for your comment. The right hemimandibles will be dissected for subsequent analysis. The tissues will be fixed in 10% buffered formalin for 24 hours, demineralized in 5% ethylenediaminetetraacetic acid (EDTA, Merck/Millipore, Burlington, MA, USA), and then processed using standard histological techniques. To improve clarity in the text, we have separated the histological and histometric analyses, as routine histological processing is a foundational step for some of the subsequent tests.

Real time reverse transcription-polymerase chain reaction

The collection of the ameloblasts is not described well enough. The methods in reference #46 are not for the pure ameloblasts collection but for enamel organs enriched with ameloblasts. Authors need to be aware of it and note it.

Authors: Thank you for your comment. First, the incisors will be carefully isolated from the mid-mandible. Next, a brief cleaning will be performed using phosphate-buffered saline (PBS 1X), followed by scraping the surface corresponding to the ameloblasts in either the secretion or maturation phase, using an excavator or a similar tool. Finally, the cervical loop of the initiator will be removed from the apical portion. The sample can then be placed in 10% formalin or a lysis solution for further investigation. These details were included in the methodology section to improve clarity and understanding. We are aware that the methods in reference #46 are not for the collection of pure ameloblasts, and this information has been included in the methodology section to enhance clarity and reproducibility.

Expected results

Compared to the amount of the experiment information, expected results are too limited. Some reports use a similar method to this protocol (reference #29, 30, and DOI: 10.2485/jhtb.16.61, etc.). Therefore, the authors should be able to present the expected results of each experiment based on the previous reports.

Authors: Thank you for your comment. Columnar ameloblasts and odontoblasts are expected to be present, along with a reduction in the thickness of both enamel and dentin (Imai et al., 2007). Additionally, it is likely that the size of the tooth germ will be diminished. In macroscopic analyses, lesions in the enamel, identified as areas of hypomineralization or opacities, are anticipated. As a result, a decrease in mineral content is expected, which will directly affect the microhardness of the tissue. Finally, a decreased expression of key peptides involved in enamel formation and maturation is expected.

Reviewer #4:

This paper highlights an important area of research addressing the effects of maternal alcohol consumption on the development of enamel. Thank you to the authors for putting this paper together and working towards helping us better understand the impact of prenatal alcohol consumption on developing dentition. Some overall concerns raised by this paper are listed below:

There are multiple tests completed in this study, with many outcome evaluations completed—photography, micro-computed tomography, Knoop microhardness test, scanning electron microscopy-energy dispersive x-ray, Raman spectroscopy, histometric analysis, immunohistochemistry, in situ zymography, real time reverse transcription-polymerase chain reaction, and statistical analysis also. It is inferred that the developmental defects of enamel (DDE) will be found in offspring from rats exposed to alcohol consumption compared to those without. With 10 different DDE analysis techniques employed, no hypothesis was provided for any of the 10 tests. Without clarity on the reasoning and anticipated findings of these tests, the study lacks focus and does not have clearly defined outcomes. It seems that the tests have been assembled in an exploratory manner without a specific, well-defined objective. A ‘Study Protocol’ should have clearly defined outcomes for the physical properties (volume, thickness, and density) of enamel and cellular changes associated with alcohol consumption.

Authors: Thank you for your comment. The presence of columnar ameloblasts and odontoblasts is expected, along with a decrease in the thickness of both enamel and dentin (Imai et al., 2007). Moreover, it is probable that the size of the tooth germ will be reduced. Macroscopic examinations are likely to reveal lesions in enamel structure, known as areas of hypomineralization or opacities. As a result, a reduction in mineral content is anticipated, which will directly influence the microhardness and density of the tissue. Finally, a decrease in the expression of key peptides involved in enamel formation and maturation is also expected.

Another concern is that the proposal does not have any content on standardizing the tests or assessing the error. Each test described in the paper needs more detailed elaboration.

Authors: The evaluation of photographs will be conducted by two assessors. The Kappa correlation coefficient will be calculated to assess agreement between them. The outcomes assessed in Micro-CT and histometric analyses will be evaluated on two occasions, with a two-week interval to determine method error. Intra-evaluator repeatability will be measured using the Intraclass Correlation Coefficient, while the presence of random and systematic errors will be assessed using the Bland-Altman method. In Knoop microhardness analyses, Raman spectroscopy, and scanning electron microscopy with energy dispersive X-ray, all samples will be discarded after the tests are completed. Finally, in histological analyses, routine procedures will be conducted with precision, ensuring that the fixative is prepared in sodium phosphate-buffered solutions and that the decalcification process is properly executed, prioritizing the use of chelating agents over acids. For adequate dehydration, the volume of alcohol used should be twenty times that of the sample. During the clarification phase, the duration in xylene will be carefully monitored to prevent the material from drying out. Impregnation will take place in an oven set to 60 degrees Celsius, and during the inclusion process, the fragments will be immersed in heated paraffin to avoid the formation of air bubbles.

Power calculation should be an essential part of ‘Study Protocol’.

Authors: Thank you for your comment. We have chosen to estimate 10 offspring of each sex per group. Considering potential losses, this results in a total of 22 offspring per group. Due to the lack of prior studies on this topic, we do not have data to calculate effect sizes for each outcome assessed, which prevents us from performing a priori sample size estimation. In this proposal, we plan to work with 88 animals and will calculate the power of the analyses a posteriori.

Statistical testing: The description on statistical testing is inadequate. Each quantitative method should contain adequate description for statistical testing. The authors should also discuss what they will do if the underlying assumptions of ANOVA or similar tests are violated.

Authors: Data will be analyzed using GraphPad Prism 8.0 (Prism, Chicago, IL, USA), with a significance level set at 5%. The relative and absolute frequencies of clinical enamel alterations observed in photographic analyses, as well as enamel mineral content assessed by SEM-EDS in both groups, will be compared using the chi-square test. Continuous data from microhardness and Raman spectroscopy analyses will be compared using one-way ANOVA to evaluate the effect of alcohol. Data from microtomographic analyses (including enamel volume, thickness, and density), histomorphometric analysis (enamel matrix area), immunohistochemistry, in situ zymography, and PCR (relative mRNA expression) will be assessed using two-way ANOVA to examine the effects of alcohol, age, and their interaction. Separate analyses will be conducted for incisors and molars. If the assumptions of the tests are not met, generalized models will be employed based on the characteristics of the data.

Additional comments:

Animals and experimental procedures

How long is the mating period? Are female rats exposed to alcohol throughout this period?

Authors: Thank you for your comment. The mating period can last approximately 5 days (Rosen et al., 1987). And yes, all females will be exposed to alcohol throughout this period, as the exposure begins in the week leading up to mating and, consequently, to gestation. This information has been emphasized in the text to enhance clarity.

How long will the rats be exposed to alcohol in total? What is done if the female rat does not become pregnant in the expected mating period?

Authors: Thanks for your comment. In total, the females will be exposed to alcohol over a period of 4 weeks. The exposure begins in the week prior to mating. After gestation is confirmed, alcohol administration will continue for 3 weeks, following the previously mentioned Binge Drinking protocol, which consists of 3 days of intervention followed by 4 days of rest (Frazão et al., 2020). If pregnancy does not occur within the expected time frame, the rat will be removed from the experiment to prevent potential alterations in the outcome due to prior alcohol exposure.

“Binge drinking is characterized by bringing BAC up to 0.08 grams % or above, having the potential to generate signs of intoxication” – what are the signs of intoxication?

Authors: Thank you for your comment. Acute levels of ethanol can impair performance on tasks related to timing, response inhibition, and position discrimination (Popke et al., 2000). Additionally, symptoms such as ataxia, drowsiness, decreased consciousness, loss of reflexes, and even unconsciousness may be observed.

Please clarify for a general reader why was a binge drinking model chosen and not a lower alcohol consumption percentage.

Authors: We selected the Binge Drinking Model because it closely resembles current alcohol consumption patterns (i.e., it reflects reality more accurately). This type of consumption is characterized by occasional and intense episodes, differing from the chronic alcoholism model, which is less common today.

3g/kg (i.e., 3gm/1,000gm x 100%) yields a value of 0.3%, and it is difficult to determine how this equates to 30% w/v.

Authors: To achieve a concentration of 30% ethanol, a dilution will be prepared using 30 ml of absolute alcohol mixed with 70 ml of distilled water. This mixture will yield the desired concentration. For dosing, the target is 3 g of ethanol per kg of body weight, which equates to 3.8 ml of ethanol per kg. Assuming each animal weighs approximately 200 g, the calculated dose will be 0.6 g of ethanol, corresponding to 0.76 ml of the prepared solution.

The specific issues with some of the methods are provided as examples below:

Photography

Some of the key data are missing, e.g., camera angulation, room lighting, magnification, and quantitative vs qualitative analysis.

Authors: Based on the literature (Schmalfuss et al., 2022) and pilot projects conducted by our research group, the animals will be positioned perpendicular to the camera lens at a 90-degree angle. For ambient lighting, we will use white light for photography. Additionally, a polarized light filter will be employed. This information has been incorporated into the text for clarity.

Micro-computed tomography

How will the authors ensure the density measurement is accurate and standardized? Will the molars and surrounding bone be removed from the Micro-CT images or from the specimens before imaging?

Authors: Thank you for your comment. To ensure the accuracy and standardization of the test, the same voltage, exposure time, and data analysis parameters will be applied to all samples: 70 kV, 200 μA, Al/Cu filter of 0. 1 mm, voxel size 5.4 μm, full circle rotation steps at 0.4° angle intervals, and average scanning time about 2 h. The specimen will undergo a cutting process to isolate the inc

---

## [Decision Letter · Decision Letter 2]

22 Nov 2024

PONE-D-24-17362R2Effect of prenatal alcohol consumption on dental enamel formation in offspring – an animal study protocolPLOS ONE

Dear Dr. de Paula-Silva,

Thank you for submitting your manuscript to PLOS ONE. After careful consideration, we feel that it has merit but does not fully meet PLOS ONE’s publication criteria as it currently stands. Therefore, we invite you to submit a revised version of the manuscript that addresses the points raised during the review process.

We look forward to receiving your revised manuscript.

Kind regards,

Boyen Huang, DDS, MHA, PhD

Academic Editor

PLOS ONE

Journal Requirements:

Reviewers' comments:

Reviewer's Responses to Questions

**Comments to the Author**

1. Does the manuscript provide a valid rationale for the proposed study, with clearly identified and justified research questions?

Reviewer #3: Yes

Reviewer #4: Yes

2. Is the protocol technically sound and planned in a manner that will lead to a meaningful outcome and allow testing the stated hypotheses?

Reviewer #3: Partly

Reviewer #4: Yes

3. Is the methodology feasible and described in sufficient detail to allow the work to be replicable?

Reviewer #3: Yes

Reviewer #4: Yes

4. Have the authors described where all data underlying the findings will be made available when the study is complete?

Reviewer #3: Yes

Reviewer #4: Yes

5. Is the manuscript presented in an intelligible fashion and written in standard English?

Reviewer #3: Yes

Reviewer #4: Yes

6. Review Comments to the Author

You may also provide optional suggestions and comments to authors that they might find helpful in planning their study.

Reviewer #3: The revision addressed most of the reviewers' concerns. However, some minor points are still required for revision.

1. Resolution of Figures: Although the new figures are saved at 600 dpi, the letters in each figure are still too blurry and unreadable. All contents and text in the Figures need to be readable.

2. Collection of ameloblasts for real-time RT-PCR: As the author is aware now, the collected sample will not be homogenous with ameloblasts but enamel organs rich in ameloblasts. Therefore, it is not appropriate to describe the "collection of ameloblasts" here, even with the remark "not specifically designed for the collection of pure ameloblasts." This remark actually reduces the reliability of the sample collection. Instead of adding the remark, it is recommended to state that the collected sample will be enamel organs rich in ameloblasts. This way, it is more accurate in describing the sample collection.

3. The model of SEM is missing. This information is necessary to be consistent with other sections including information on all instruments' makes and models.

Reviewer #4: Overall the paper has greatly improved in quality since the previous review, thank you to the authors for their hard work.

There are still areas in which can be improved, mostly around grammar. The paper requires a few more read-throughs by the authors and minor edits for comprehension. For example “having the potential to

generate signs of intoxication, impairing performance on tasks related to timing, response

inhibition, and position discrimination” on page 6, and “… we will take photographs of right hemimandibles from both groups (n=22), along with incisors and molars (n=22) from the left hemimandible in each group” on page 10. The wording of both these examples can be further improved.

Please also review paper for finer details – for example on page 6, “It differs from the chronic alcoholism model, which is less common today” – no definition of the chronic alcoholism model is provided, this comparison is consequently unclear. Additionally on page 14, “… secondary antibody for 1 hour, washed in PBS” – The acronym for PBS has not been introduced. Please review full article to further improve readability.

7. PLOS authors have the option to publish the peer review history of their article (what does this mean?). If published, this will include your full peer review and any attached files.

Reviewer #3: No

Reviewer #4: No

---

## [Author Response · Author response to Decision Letter 2]

10 Dec 2024

Reviewer #3:

Authors: Thank you for your comment. For the SEM-EDS, RT-RAMAN, and in situ zymography outcomes, due to the lack of previous studies, these are considered exploratory analyses, and it is not possible to precisely define the expected results. Furthermore, the methods used for each procedure, such as the collection of ameloblasts, may be modified or improved as the research progresses. This information has been added to the text and is highlighted in red.

Resolution of Figures: Although the new figures are saved at 600 dpi, the letters in each figure are still too blurry and unreadable. All contents and text in the Figures need to be readable. 

Authors: Thank you for your comment. The figures have been revised and adjusted on the PACE platform according to the specifications of PLOS ONE.

Collection of ameloblasts for real-time RT-PCR: As the author is aware now, the collected sample will not be homogenous with ameloblasts but enamel organs rich in ameloblasts. Therefore, it is not appropriate to describe the "collection of ameloblasts" here, even with the remark "not specifically designed for the collection of pure ameloblasts." This remark reduces the reliability of the sample collection. Instead of adding the remark, it is recommended to state that the collected sample will be enamel organs rich in ameloblasts. This way, it is more accurate in describing the sample collection.

Authors: Thank you for your comment. The information in the text has been reviewed, and the requested correction has been made.

The model of SEM is missing. This information is necessary to be consistent with other sections including information on all instruments' makes and models.

Authors: Thank you for your comment. The model has been added (JEOL JSM-6610LV) and is highlighted in red.

Reviewer #4:

Overall, the paper has greatly improved in quality since the previous review, thank you to the authors for their hard work. There are still areas in which can be improved, mostly around grammar. The paper requires a few more read-throughs by the authors and minor edits for comprehension. For example, “having the potential to generate signs of intoxication, impairing performance on tasks related to timing, response inhibition, and position discrimination” on page 6, and “… we will take photographs of right hemimandibles from both groups (n=22), along with incisors and molars (n=22) from the left hemimandible in each group” on page 10. The wording of both these examples can be further improved. 

Authors: Thank you for your comment. The entire grammar of the project has been revised, and the necessary adjustments have been made. We appreciate the suggestion.

Please also review the paper for finer details – for example on page 6, “It differs from the chronic alcoholism model, which is less common today” – no definition of the chronic alcoholism model is provided, this comparison is consequently unclear. Additionally on page 14, “… secondary antibody for 1 hour, washed in PBS” – The acronym for PBS has not been introduced. Please review the full article to further improve readability.

Authors: Thank you for your comment. The entire article has been reviewed to further enhance readability.

---

## [Decision Letter · Decision Letter 3]

2 Jan 2025

Effect of prenatal alcohol consumption on dental enamel formation in offspring – an animal study protocol

PONE-D-24-17362R3

Dear Dr. de Paula-Silva,

We’re pleased to inform you that your manuscript has been judged scientifically suitable for publication and will be formally accepted for publication once it meets all outstanding technical requirements.

Kind regards,

Boyen Huang, DDS, MHA, PhD

Academic Editor

PLOS ONE

Additional Editor Comments (optional):

Reviewers' comments:

Reviewer's Responses to Questions

**Comments to the Author**

1. Does the manuscript provide a valid rationale for the proposed study, with clearly identified and justified research questions?

Reviewer #3: Partly

Reviewer #4: Yes

2. Is the protocol technically sound and planned in a manner that will lead to a meaningful outcome and allow testing the stated hypotheses?

Reviewer #3: Partly

Reviewer #4: Yes

3. Is the methodology feasible and described in sufficient detail to allow the work to be replicable?

Reviewer #3: Yes

Reviewer #4: Yes

4. Have the authors described where all data underlying the findings will be made available when the study is complete?

Reviewer #3: Yes

Reviewer #4: Yes

5. Is the manuscript presented in an intelligible fashion and written in standard English?

Reviewer #3: Yes

Reviewer #4: Yes

6. Review Comments to the Author

You may also provide optional suggestions and comments to authors that they might find helpful in planning their study.

Reviewer #3: This revision addressed most of the comments from the reviewer. However, the texts in the figures are still not readable enough on my end in the review package.

Reviewer #4: All comments noted in previous reviews have been addressed by the authors. Thank you again for your hard work in this paper.

7. PLOS authors have the option to publish the peer review history of their article (what does this mean?). If published, this will include your full peer review and any attached files.

Reviewer #3: No

Reviewer #4: No

---

## [Editor Report · Acceptance letter]

5 Jan 2025

PONE-D-24-17362R3 

PLOS ONE

Dear Dr. de Paula-Silva, 

I'm pleased to inform you that your manuscript has been deemed suitable for publication in PLOS ONE. Congratulations! Your manuscript is now being handed over to our production team.

Kind regards, 

on behalf of

Dr Boyen Huang 

Academic Editor

PLOS ONE